# Multimorbidity Management: A Scoping Review of Interventions and Health Outcomes

**DOI:** 10.3390/ijerph22050770

**Published:** 2025-05-13

**Authors:** Kagiso P. Seakamela, Reneilwe G. Mashaba, Cairo B. Ntimana, Chodziwadziwa W. Kabudula, Tholene Sodi

**Affiliations:** 1Department of Pathology and Medical Sciences, Faculty of Health Sciences, School of Medicine, University of Limpopo, Sovenga, Polokwane 0727, South Africa; cairo.ntimane@ul.ac.za; 2Dikgale Mamabolo Mothiba (DIMAMO) Population Health Research Centre, University of Limpopo, Sovenga, Polokwane 0727, South Africa; given.mashaba@ul.ac.za; 3MRC/Wits Rural Public Health and Health Transitions Research Unit (Agincourt), Faculty of Health Sciences, School of Public Health, University of the Witwatersrand, Johannesburg 2193, South Africa; chodziwadziwa.kabudula@wits.ac.za; 4Department of Psychology, University of Limpopo, P/Bag X1106, Sovenga, Polokwane 0727, South Africa; tholene.sodi@ul.ac.za

**Keywords:** multimorbidity, interventions, multidisciplinary approaches, outcomes, management

## Abstract

Multimorbidity, defined as the co-occurrence of two or more chronic conditions in an individual, has emerged as a worldwide public health concern contributing to mortality and morbidity. This complex health phenomenon is becoming increasingly prevalent worldwide, particularly as populations continue to age. Despite the growing burden of multimorbidity, the development and implementation of interventions published by scholars are still in their early stages with significant variability in strategies and outcomes. The variability in strategy and outcome may result from factors such as lack of infrastructure, socioeconomic status and lifestyle factors. The review aims to synthesize interventions designed to manage and mitigate multimorbidity and explore a range of approaches, including pharmacological treatments, lifestyle modifications, care coordination models, and technological innovations. The scoping review was conducted following the Preferred Reporting Items for Systematic Reviews and Meta-Analyses extension for Scoping Reviews (PRISMA-ScR) Checklist. It included 1,553,877 individuals with multimorbidity with no age restriction; in the studies that included gender difference, 463,339 male participants and 1,091,538 female participants were involved. Multimorbidity interventions were defined as strategies or programs designed to manage and improve the health and quality of life of individuals with multiple chronic conditions. Of the downloaded articles, those that met the inclusion criteria were published between 2012 and 2024. The final analysis included 100 articles from 3119 published articles, which resulted in 9 themes and 15 subthemes. Themes on the need for lifestyle and behavioural interventions, patient empowerment and engagement, multimorbidity management, health integration, pharmacotherapy optimization, community and policy interventions, healthcare system improvements, technology and digital health, as well as research and evidence-based practice interventions, emerged. The reviewed literature emphasizes the necessity of multidisciplinary approaches to effectively combat the growing public health challenge of multimorbidity.

## 1. Introduction

Multimorbidity has become a global public health challenge that contributes to both mortality and morbidity [1,2,3]. This complex health phenomenon is increasingly becoming prevalent (37.2% globally), affecting populations especially as they age [4,5]. Multimorbidity complicates clinical management and exacerbates health outcomes, leading to reduced quality of life, and increased health utilization and costs [6,7,8]. Even though the prevalence of multimorbidity is projected to remain on an upward trend, policy and research remain focused on single diseases, making it difficult to prevent multimorbidity better [9,10].

Multimorbidity has been described as a “defining challenge” for health systems, which are traditionally focused on single conditions [11]. This calls for effective interventions to address the multifaceted needs of patients with multimorbidity, encompassing a holistic approach that integrates medical, psychological, and social dimensions of care [12,13,14]. Despite the growing burden of multimorbidity, the development and implementation of interventions published by scholars are still in their early stages with significant variability in strategies and outcomes [12,13,14]. In addition, the application of the proposed interventions may be affected by the socioeconomic status of the respective countries.

The vast research published on multimorbidity recommends possible interventions that scientists and policymakers could adopt [7,15,16]. This review aims to synthesize interventions designed to manage and mitigate multimorbidity and explore a range of approaches, including pharmacological treatments, lifestyle modifications, care coordination models, and technological innovations.

### Background on Multimorbidity Interventions

Multimorbidity interventions are defined as strategies or programs designed to manage and improve individuals’ health and quality of life with multiple chronic conditions [15]. Given the challenge of managing several chronic illnesses simultaneously, multimorbidity management has placed a greater emphasis on patient-centred interventions. Integrated care approaches, which promote collaboration across interdisciplinary teams to address patients’ medical, social, and psychological needs holistically, are among the most crucial strategies. Patient empowerment via self-management measures, such as education on medication adherence and lifestyle changes, improves results and lessens the burden of therapy. Remote monitoring and telemedicine are two examples of technology-based therapies that have demonstrated potential in improving accessibility and continuity of care, especially for underprivileged populations. To make sure that therapies meet patients’ objectives and lower the hazards of polypharmacy (such as adverse drug interactions), these interventions are supplemented by shared decision-making frameworks and systematic medication reviews [13,15,17].

## 2. Materials and Methods

This scoping review was conducted and reported following the Preferred Reporting Item for Systematic reviews and Meta-Analyses extension for Scoping Reviews (PRISMA-ScR) Checklist (see Appendix A) [18]. To ensure transparency, the authors searched multiple databases for similar scoping reviews to avoid duplication. Without such a review, the authors continued with the scoping review. The protocol for this review was not registered.

### 2.1. Search Strategy

To reduce selection bias, three (3) authors (K.P.S., R.G.M., and C.B.N.) systematically searched for relevant studies from inception to June 2024. The search strategy was performed between 1 April 2024 to 30 June 2024. Five electronic databases (PubMed, Google Scholar, Science Direct, Cochrane and Scopus) were searched for relevant evidence. The search strategy was first drafted and tested using PubMed. This was further adapted to the syntax and subject headings of all other databases searched in the study. Keywords used in the search were “Multimorbidity”, “Multiple chronic conditions”, and “Interventions”. In addition, reference lists of all eligible articles identified were searched and screened for additional relevant studies. The search strategy for PubMed has been included in the Appendix A as an example.

### 2.2. Inclusion and Exclusion Criteria

Studies were reviewed against pre-determined inclusion and exclusion criteria for eligibility. Peer-reviewed studies were included if they investigated multimorbidity interventions or proposed multimorbidity interventions in their recommendations, written in English and not reviews. Studies were excluded if they focused on single morbidity, did not have multimorbidity interventions, were not written in English and were reviewed. The present review did not have age and type of multimorbidity restrictions.

### 2.3. Study Selection

Selection and inclusion of papers for this review involved a two-stage process: screening of abstracts and titles; and full-text reading to select eligible papers for final inclusion. Three independent reviewers (K.P.S., R.G.M., and C.B.N.) conducted the selection process through each stage of the review. After the database search, the results were exported to Zotero version 6.0, a bibliographic management software, where duplicates were removed. After removing the duplicates, the reviewers applied the pre-determined inclusion and exclusion criteria and independently assessed the titles and abstracts for full-text review eligibility. Following this process, articles were selected for full-text screening. Again, the reviewers applied the inclusion and exclusion criteria and independently assessed the full-text articles. After each stage of the selection process, the reviewers compared results and reached a consensus, with a fourth reviewer and a fifth reviewer serving as a tie-break in situations where the three reviewers failed to reach an agreement.

### 2.4. Data Extraction and Analysis

Relevant data were extracted from eligible articles using a Microsoft Excel spreadsheet. These included author name, date of publication, country of study, the main objective of the study, study design and participants, sample size and gender distribution. The research team quantified the characteristics of the included research and tabulated important data. The study findings were then thematically synthesized. Thereafter, the research team created a thematic framework by discussing the thematic codes. K.P.S., C.B.N., and R.G.M. then applied the remaining articles and verified that all the coding was accurate. K.P.S. and R.G.M. quantified codes using NVivo version 14-word clouds. After that, researchers created themes that were approved by all the reviewers. R studio’s (version 4.4.0) UpSetR and ggplot2 libraries were used to generate an upSet plot.

## 3. Results

Out of a pool of 3119 articles identified from the database, full-text articles were assessed for eligibility. About 1400 articles were excluded due to not being relevant to the present review (*n* = 280), lack of multimorbidity interventions (*n* = 868), focus on single morbidity (*n* = 196) and reviews (*n* = 56) (Figure 1). The final analysis included 101 articles from 3119 published between 2012 and 2024. The included studies comprised various research designs: 55 randomized controlled trials (53.5%), 32 cross-sectional (32.7%) and 6 cohorts (5.9%). Longitudinal, qualitative and population-based studies contributed about 2.9%, respectively. The sample sizes of these studies ranged from 6 to 712,822 participants.

In total, 1,553,877 individuals with multimorbidity participated in these studies, with 463,339 being male and 1,091,538 females. Three articles did not report the sample size, while twelve studies reported the total sample size but not gender distribution. The geographical distribution of these studies was as follows: about 27 studies were from Africa, 5 Asia, 2 Australia, 30 Europe and 15 North America. About 21 of the included studies were multinational studies. Figure 2 presents percentages of studies from each continent and percentages of individual countries within respective continents. Figure 3 presents the distribution of interventions in the 101 studies included in the review, stratified by countries’ financial status. The patterns of recommended interventions included patient empowerment, engagement and pharmacotherapy optimization, lifestyle and behavioural interventions as well as healthcare system improvement. These interventions were higher in low-income compared to high-income countries. In high-income countries, the interventions included mental health integration as an individual pattern.

Furthermore, Appendix A presents a summary of the included studies’ characteristics.

Upon thematic analysis, this scoping and review identified 9 themes, which are summarized in Table 1.

**Table 1 ijerph-22-00770-t001:** Summary of themes identified.

Themes	Subthemes
Lifestyle and Behavioural Interventions	1.1. Physical activity promotion and exercise programmes (aerobic and resistance) [19,20,21,22,23,24,25,26,27,28,29,30] 1.2. Smoking cessation, alcohol reduction and weight management [19,23,27,28,31,32,33,34,35,36]
2. Patient Empowerment and Engagement	2.1. Self-management skills training [21,26,37,38,39,40,41,42,43,44,45]. 2.2. Assisting patients with goal setting and prioritization, peer support, health service navigation [26,27,43,46].
3. Collaborative and Patient-Centered Care	3.1. Multidisciplinary and multi-sectoral collaboration and support in healthcare systems [47,48,49,50]. 3.2. Patient-centred care [51,52,53,54,55]. 3.3. Preventive strategies for chronic disease and modifiable risk factors [24,30,47,48,56,57,58,59,60,61,62,63].
4. Mental Health Integration	4.1. Integrating mental health screening and treatment along with collaborative care for mental health and physical conditions [26,27,30,37,64,65,66,67,68,69].
5. Pharmacotherapy Optimization	5.1. Medication Reviews [38,40,50,70,71,72,73,74]. 5.2. Use of software-based tools and clinical decision support systems [70,75,76,77,78]. 5.3. Shared decision-making with patients [75,78,79,80,81].
6. Community and Policy Interventions	6.1. Policy interventions for physical activity infrastructure and community engagement programs [20,26,44,48,82,83,84]. 6.2. Financial assistance [20,35,85,86].
7. Healthcare System Improvements	7.1. Enhancing communication between primary care and hospitals [26,39,40]. 7.2. Strengthening capacity and training for healthcare workers [26,39,40].
8. Technology and Digital Health	[48,68,87,88,89].
9. Research and Evidence-Based Practice	[6,31,34,35,52,57,63,90,91,92,93,94,95,96,97,98,99].

### 3.1. Lifestyle and Behavioural Interventions

#### 3.1.1. Physical Activity

Lifestyle changes were identified as a positive factor in managing multimorbidity. About eleven (11) studies indicated that physical activity and exercise programs were important among individuals with multimorbidity [19,21,22,23,24,25,26,27,28,29,30]. This included encouraging individuals with multimorbidity to engage in regular exercise, which may be incorporated into community programs. In addition, the provision of individual counselling and access to fitness facilities and organized activities were noted to have potential benefits for individuals with multimorbidity. Designing structured exercise programs tailored for individual capabilities and extended multimorbidity was also highlighted. Some studies suggested cost-effective exercise programs consisting of aerobics [21,29] and low-impact cycling to improve body composition [82]. Participating in brisk walking for more than 175 minutes per week also had a positive impact on inducing weight loss [25].

#### 3.1.2. Smoking Cessation, Alcohol Reduction and Weight Management

Lifestyle interventions such as quitting smoking, healthy diet and alcohol reduction were found in several studies to have a positive impact on the management of multimorbidity [19,23,27,28,31,32,33,34]. Interventions such as offering support and resources for quitting smoking, implementing weight management programs that include dietary advice and physical activity, programs to reduce alcohol consumption and community-based initiatives to promote overall wellness were suggested. One study reported that reducing caloric intake and increasing physical activity to induce weight loss to average >7% at year 1 and maintaining this over time could lessen the burden of multimorbidity [25].

### 3.2. Patient Empowerment and Engagement

#### 3.2.1. Self-Management Skill Training

Several studies reported the importance of teaching patients’ skills to manage individual chronic health conditions independently and offering programs that build confidence and competence in self-care and self-monitoring to adjust lifestyle habits and make healthy lifestyle choices [21,26,37,38,39,40,41,42,43,44,45]. Mbokazi et al. further highlighted the need to incorporate and recognize cultural norms in interventions, including the role that Ubuntu plays for people living with multimorbidity in South Africa and the need to consider how existing social networks can be strengthened to enhance self-management capacity [44]. This is especially true in a changing healthcare landscape where patients are expected to take more responsibility for their health. Tomita et al. further recommended a restructure of the social welfare framework and establishment of culturally sensitive healthcare systems capable of managing multimorbidity amidst the unprecedented urbanization occurring in Tanzania [100]. Information for patients to increase self-management abilities and reduce language barriers and difficulties of comprehension was found useful in achieving this intervention [38,101]. In addition, addressing key non-disease-specific self-management challenges and themes (mastering emotions, managing treatments, and communication within healthcare) may assist in patient self-management [39]. The following suggestions were also highlighted: developing accessible educational resources for patients on various health topics and providing printed and digital materials to support patient learning; organizing group sessions where patients can learn and practice self-management skills; facilitating peer support and sharing of experiences.

#### 3.2.2. Assisting Patients with Goal Setting and Prioritization, Peer Support, Health Services Navigation

It is important to help patients set realistic and achievable health goals and prioritize interventions based on what is important to the patient [26,27,46]. Goal setting and prioritization based on patient preferences should include interventions to support patient self-management used in groups including self-management, fatigue and energy management, managing stress, anxiety, and maintaining mental health and well-being, keeping physically active, healthy eating, managing medications, effective communication strategies, and goal setting [26]. In addition, facilitating peer support groups where patients can share experiences and advice was found to be necessary for patient empowerment and engagement [26]. Furthermore, Garvey et al. pointed out the importance of helping patients with multimorbidity to navigate the healthcare system and provide them with guidance on how to access services and coordinated care by employing health navigators’ services [26]. Health navigators are professionals who can assist individuals with multimorbidity to overcome factors that may prevent them from accessing healthcare (e.g., financial, transportation, knowledge-based skills, and complex medical systems) to help them manage and cope with disease-related symptoms and health problems [26]. 

### 3.3. Collaborative and Patient-Centered Care

#### 3.3.1. Multidisciplinary and Multi-Sectoral Collaboration and Support in Healthcare Systems

The importance of establishing integrated care pathways that connect various healthcare providers and services as well as engaging social services, community organizations, and policymakers in comprehensive care plans has been highlighted [47,48,49,50]. This ensures continuity of care across different settings and specialities and facilitates teamwork among healthcare professionals from various disciplines, thus encouraging shared decision-making and team-based care [47,48,49,50]. Multi-sectoral interventions are required beyond the healthcare sector to reduce the impact of NCDs on health systems and broader societal development [49].

#### 3.3.2. Patient-Centred Care

Several studies suggested focusing on the individual needs and preferences of patients and implementing personalized care plans that address multiple health conditions. In addition, restructuring healthcare delivery models to be more efficient and patient-centred and implementing care teams and case management approaches have been highlighted [51,52,53,54,55]. The current models of health care delivery need to be re-examined, and patient-centred models of integration need to be evaluated, including bidirectional screening of commonly co-morbid conditions in routine clinical practice [52]. A cost-effective and patient-centred approach to the treatment and care of multiple comorbidities is required in treating multimorbidity [51,54].

#### 3.3.3. Preventive Strategies for Chronic Disease and Modifiable Risk Factor

Several studies highlighted the need for prevention and early intervention for chronic diseases including regular monitoring and management to prevent complications [30,48,58,60,61]. This includes prevention initiatives tailored to non-communicable diseases with a special target of diseases and health conditions that have the most detrimental effects on health-related quality of life [60]. Affordable, broad-scale interventions, such as those promoting healthy lifestyles and preventing and managing chronic conditions, and subsequently, multimorbidity [30], are needed. In addition, other studies reported the need to increase activities and expand measures to reduce the modifiable risk factors that are driving multi-morbidity prevalence [25,47,48,56,57,99]. Concerted efforts to develop strategies for the planning, prevention and management of modifiable risk factors that drive the high prevalence of multimorbidity including monitoring of cardiometabolic risk factors (lipids, HbA1c, blood pressure) are needed [25,57].

### 3.4. Mental Health Integration

#### Integrating Mental Health Screening and Treatment Along with Collaborative Care for Mental Health and Physical Conditions

Implementing models of care that integrate mental health services with primary care using team-based approaches to manage co-existing mental and physical health conditions is vital [26,27,30,37,64,65,66,67,68]. This is because mental health conditions tend to affect the treatment outcomes of patients with multimorbidity. This can be achieved by providing access to therapies like cognitive-behavioural therapy (CBT) and encouraging patients to track their mental health symptoms and triggers. Routine screening for depression and anxiety in patients with multimorbidity and offering treatment options that address both mental and physical health needs are recommended. Oh et al. reported the need for interventions aimed at minimizing distress, which has been associated with a lower adherence to treatment, leading to worsening of disease severity and outcomes. Raising awareness about distress and related mental health problems and promoting the utilization of mental healthcare among people with multimorbidity are highly important [102].

### 3.5. Pharmacotherapy Optimization

#### 3.5.1. Medication Reviews

Several studies highlighted the importance of regular comprehensive reviews of patient medications to ensure appropriateness, effectiveness, and safety with the aims of identifying and discontinuing potentially inappropriate medications to reduce significant polypharmacy by de-prescription of unnecessary medicines [38,40,50,70,71,72,73]. Medication reviews also increase expert knowledge and feasibility of instruments for systematic medication reviews, and implementation of an integrated healthcare model for multimorbid patients based on improving communication between primary care and hospital professionals [38,40,50,70,71,72,73]. Additionally, the studies highlighted the need to perform a brown bag medication review, record any problems identified and assess and record patients’ priorities for treatment [38,40,50,70,71,72,73].

#### 3.5.2. Use of Software-Based Tools and Clinical Decision Support Systems

Utilizing tools like STOPP (Screening Tool for Older People’s Prescriptions) and START (Screening Tool to Alert to the Right Treatment) criteria to guide prescribing practices and implementing clinical decision support systems that incorporate these criteria are critical [70,75,76,77,78]. This assists in reducing inappropriate prescribing through systematic medication and predicting adverse medication effects, advising safe and appropriate therapy using and monitoring clinically relevant interactions and dosing [47]. Factors such as Structured History taking of Medication use (SHiM) and collection of patient data including medical conditions, laboratory data and clinical parameters and digitalization of screening of pharmacotherapy through a Clinical Decision Support System (CDSS) should be incorporated to detect potential overuse, underuse, and misuse of drugs [101]. The use of software-based tools and clinical decision support systems assist in improvements in medication reconciliation across healthcare settings to avoid unintentional re-prescription of medication [70,75,76,77,78].

#### 3.5.3. Shared Decision-Making with Patients

Involving patients in discussions about their medications, considering their preferences and experiences, is essential [75,78,79,80]. This also includes educating patients about the benefits and risks of their treatments.

### 3.6. Community and Policy Interventions

#### 3.6.1. Policy Interventions for Physical Activity Infrastructure and Community Engagement Programs

About seven (7) articles reported the need to develop public health policies that promote the construction of recreational facilities to encourage and facilitate physical activity [20,26,44,48,82,83,84]. In addition, the importance of community engagement events and programs that promote health and well-being and foster social connections and support networks were highlighted as probable interventions [20,26,44,48,82,83,84]. Furthermore, the studies highlighted the need to address social determinants of health through community-based approaches [20,26,44,48,82,83,84].

#### 3.6.2. Financial Assistance

Factors such as lack of financial means were also found to play a role in the management or lack thereof of multimorbidity. As an intervention, some studies therefore recommended the need to offer financial support for individuals with multimorbidity to eliminate the financial barrier they face in accessing health services [20,35,85].

### 3.7. Healthcare System Improvements

#### 3.7.1. Enhancing Communication Between Primary Care and Hospitals

The present review found that there remained a lack of communication between healthcare providers in hospitals, which needs to be improved through the incorporation of health information technology and integrated care systems that connect primary, secondary, and tertiary care to facilitate communication between health professionals and enhance coordination between various levels of care [26,39,40].

#### 3.7.2. Strengthening Capacity and Training for Healthcare Workers

Providing ongoing education and training for healthcare workers for multimorbidity management is necessary. This includes offering specialized training for healthcare providers on managing multimorbidity and conducting workshops and seminars to enhance clinical skills [22,38,84,91,103].

### 3.8. Technology and Digital Health

Some studies revealed the need for technology and digital health in the management of multimorbidity [48,68,87]. This includes internet-based self-management programs, telehealth and telephone nursing support, electronic registries and tracking systems and the use of mobile apps for patient engagement [48,68,87]. Some studies promoted the use of technology and digital health to offer patients online platforms to learn about and manage multimorbidity [48,68,87]. This further provides healthcare workers with the opportunity to track and monitor health metrics, remote consultations and follow-ups, and monitor health outcomes. Creating mobile applications that support patient engagement and self-management was also highlighted [48,68,87].

### 3.9. Research and Evidence-Based Practice

Conducting long-term research to investigate underlying mechanisms and risk factors to understand the causes and risk factors for multimorbidity, exploring the interactions between different health conditions, tracking the progression and impact of multimorbidity and analyzing data to inform future healthcare practices are vital [6,31,34,35,52,57,63,90,91,92,93,94,95,96,97]. This includes creating and testing new approaches to managing multimorbidity, focusing on interventions that are patient-centred and effective [6,31,34,35,52,57,63,90,91,92,93,94,95,96,97], assessing the effectiveness of integrated care models and using evidence to refine and improve care delivery systems [6,31,34,35,52,57,63,90,91,92,93,94,95,96,97].

Figure 4 presents the interrelation between interventions. Healthy lifestyle changes and behavioural interventions improve mental well-being. Patients with better mental health feel more empowered to actively participate in managing multimorbidity at a personal level and participate in collaborative care. Better digital health solutions for monitoring and patient engagement are needed, which enable pharmacotherapy optimization, ensuring that medications are effectively tailored to patient needs.

## 4. Discussion

This review aimed to synthesize interventions designed to manage and mitigate multimorbidity and explore a range of approaches, including pharmacological treatments, lifestyle modifications, care coordination models, and technological innovations. The results of the current review revealed the need for lifestyle and behavioural interventions, patient empowerment and engagement, multimorbidity management, health integration, pharmacotherapy optimization, community and policy interventions, healthcare system improvements, technology and digital health, as well as research and evidence-based practice interventions. The reviewed literature emphasizes the necessity of multidisciplinary approaches to effectively combat the escalating pandemic of multimorbidity.

The reviewed studies suggest the need for lifestyle changes that include physical activity targeting weight loss as well as reducing alcohol consumption and smoking cessation as a mechanism that can improve multimorbidity management [19,21,22,23,24,25,26,27,28,29,30,102,104,105,106]. Smoking, alcohol consumption and physical inactivity are well-known risk factors for multimorbidity [107,108]. Since the majority of people suffering from multimorbidity are older, the literature recommends exercises that include aerobics and brisk walking, which have been reported to improve general body weight [39,109]. A study by Zou and colleagues reported smoking behaviour, age of initiation, frequency of daily smoking and passive smoking to increase the risk of multimorbidity [110]. The increased risk of multimorbidity among smokers can be explained by the reported implication of smoking towards the individual impact of smoking on cardiovascular risk including diabetes, heart disease and stroke morbidities [110,111,112].

It is worth noting that self-care plays a crucial role in the successful management of chronic diseases and multimorbidity. In the reviewed literature, patient empowerment and engagement were highlighted as important recommendations to combat the burden of multimorbidity [21,26,37,38,39,40,41,42,43,44,45,76,113,114,115]. Financial and educational status have been associated with an increased risk of multimorbidity; therefore, these two factors impact how patients suffering from multimorbidity manage their conditions [6,116,117]. Interventions aimed at overcoming the financial and educational barriers emerged from the reviewed literature where it was suggested that patients should be provided with culturally sensitive materials and educated on chronic conditions and management, patient-centred goal setting, and peer support [21,26,37,38,39,40,41,42,43,44,45]. Empowering patients with these skills and having peers who advocate for effective self-care will bring confidence in patients with multimorbidity, which will help them navigate the healthcare system. Also, the shared decision-making with patients about their medication, preferences and experiences could be beneficial to the healthcare system in pursuit of successful multimorbidity management [75,78,79,80]. A scoping review by Marzban and colleagues managed to show a positive impact of patient engagement on treatment adherence and self-care [118].

Multimorbidity is a complex phenomenon that requires more than the traditional approaches to healthcare provision. The findings of this review suggest multidisciplinary and multi-sectoral collaboration and support in healthcare systems [47,48,49,50]. The involvement of various healthcare providers in patient care would be beneficial to fragmented healthcare systems in that there would be shared patient-care decision-making. There is also a need for patient-centred care that serves the needs of individuals as patients are different. Studies have shown the beneficial impact of multidisciplinary approaches in patient care [119,120]. Multimorbidity has been associated with mental health conditions; this can be due to the observed reduction in quality of life, medication side effects and fear of death [121,122,123]. Multidisciplinary approaches aimed at integrating mental health into primary healthcare and facilitating pharmacotherapy optimization are necessary in caring for patients with multimorbidity according to the reviewed literature [26,27,28,30,37,64,65,66,67,68,90,124,125,126].

The reviewed literature suggests that medication reviews and the use of software-based tools and clinical decision support systems can achieve pharmacotherapy optimization [38,40,50,70,71,72,73,101,127]. Patients with multimorbidity are often subjected to polypharmacy; the reviewed literature recommends medication reviews to reduce significant polypharmacy by de-prescription of unnecessary medicines. Models such as STOPP, START, SHiM and CDSS are reported to be promising interventions to reduce polypharmacy and monitoring prescriptions. Software-based tools and clinical decision support systems assist in improvements in medication reconciliation across healthcare settings to avoid unintentional re-prescription of medication [70,75,76,77,78].

The results of this review revealed the need for improvement in the healthcare system. This includes continuity of care and better communication between healthcare providers (primary healthcare providers and hospitals). Suggested interventions include the development of a digital system that will link the information of the patient across all platforms of care, internet-based self-management programs, telehealth and telephone nursing support, electronic registries and tracking systems and the use of mobile apps for patient engagement [48,68,87]. Implementing these systems would reduce the need to go to healthcare facilities and queue for hours in some areas, and the cost of travelling to and from the health facilities, which are obstacles to people in extreme poverty in accessing healthcare. Also, the digital systems give an overview and perspective of the longitudinal health of a patient, which has been reported to improve patient care [128,129,130,131,132].

Strengthening capacity and training for healthcare workers through ongoing specialized training on managing multimorbidity is also recommended by the reviewed literature, which is essential in medication regimens and the application of technology. The continued training will allow healthcare providers and health systems to manage multimorbidity, which is complex and needs different approaches than the ones already being used [133,134,135].

Patterns of recommended interventions including patient empowerment and engagement and pharmacotherapy optimization, lifestyle and behavioural interventions as well as healthcare system improvement were higher in low-income compared to high-income countries. Due to financial constraints, healthcare systems in low-income nations are inadequate when compared to those in high-income nations. Compared to low-income nations, high-income countries generally invest more in the upkeep and optimization of their infrastructure. Low-income nations’ insufficient healthcare systems are largely caused by geographic limitations and a lack of medical professionals [136,137]. The inadequate healthcare systems perpetuate the lifestyle and behavioural interventions in low-income countries due to the unavailability of expensive medical interventions [138]. Patient empowerment and engagement paired with pharmacotherapy optimization were high in low-income countries compared to high-income countries. To achieve pharmacotherapy optimization, there is a need for patient empowerment and engagement that will involve patient-centered care and shared decision-making, which positively impact adherence to care. This would be nearly impossible in low-income countries if the socioeconomic status and healthcare funding issues were not resolved.

The present review has found an interrelationship between the respective interventions. For instance, healthy lifestyle changes and behavioural interventions were found to improve mental well-being and subsequent management of multimorbidity. These findings are in agreement with studies where factors such as frequent physical activity, improved diet and psychosocial interventions are reported to improve mental health (e.g., decrease depressive symptoms). This ultimately improves self-management of multimorbidity at a personal level [137,139]. Individuals with better mental health were found to participate in collaborative care initiatives. Furthermore, better digital health solutions for monitoring and patient engagement were found to enable pharmacotherapy optimization, which ensures that medications are effectively tailored to patient needs.

There is a need for long-term research to investigate underlying mechanisms and risk factors to understand the causes and risk factors of multimorbidity. Therefore, the data generated from long-term investigations can be used to create evidence-based policies that effectively serve the communities and reduce the burden of multimorbidity and chronic conditions [34,93,94,95,96,97].

### Strengths and Limitations

The review included different study types/designs, making it possible to have a comprehensive overview of multimorbidity interventions. By including studies from different locations based on countries’ socioeconomic status (low, low–middle, and high-income countries) the review uncovered the feasibility of interventions in different regions. The determination of intervention patterns gives policymakers an approach to curb the complexity of multimorbidity management by implementing tailored and evidence-based approaches to apply in their specific locations. Differences in how studies measured multimorbidity outcomes complicated the synthesis of findings and intervention effectiveness.

Overall, not all studies that were included in the review reported gender differences. Of those that reported gender differences, the majority consisted mostly of females compared to males.

## 5. Conclusions

This review underscores the growing complexity of multimorbidity and the urgent need for innovative and comprehensive approaches to manage it effectively. The synthesis of current interventions highlights the critical importance of lifestyle and behavioural changes, patient empowerment, and a more integrated healthcare system. It is worth noting that intervention effectiveness depends on healthcare systems. By applying the recommended multidisciplinary care models, optimizing pharmacotherapy, and leveraging technology, we can improve the quality of life for those living with multiple chronic conditions. Moreover, the review revealed the value of personalized care, where the unique needs and circumstances of individuals are acknowledged and addressed.

Promoting self-care and treatment adherence requires patient empowerment via education and collaborative decision-making, particularly in regions with limited resources and educational opportunities. As we move forward, the advancement of digital health technologies and the inclusion of mental health services in primary care will be essential in tackling the complex issues surrounding multimorbidity in the future. Effective multimorbidity management requires a team effort involving legislators, patients, communities, and healthcare practitioners to build a more adaptable and durable healthcare system. There is a need for the development of uniform criteria for evaluating multimorbidity interventions that could enhance comparability across studies. In terms of research, more studies are needed in resource-limited settings to evaluate the effectiveness of multimorbidity interventions in diverse healthcare systems. Reducing polypharmacy by introducing single-pill combinations for common age-related diseases is another intervention that should be explored.

## Figures and Tables

**Figure 1 ijerph-22-00770-f001:**
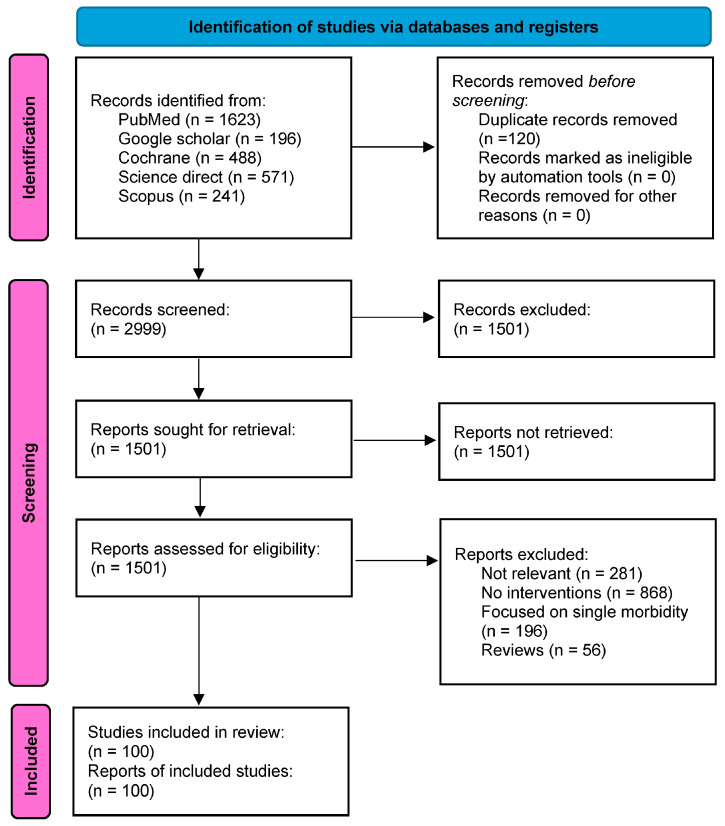
Study selection flow diagram.

**Figure 2 ijerph-22-00770-f002:**
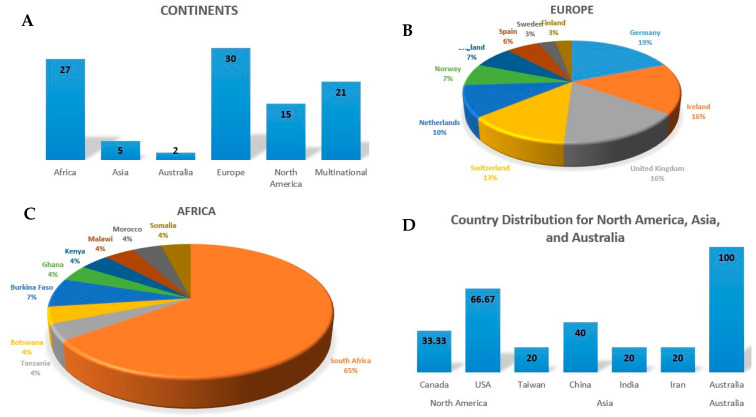
Distribution of countries and continents from which included studies were conducted. (**A**): The continental distribution with Europe and Africa demonstrating the highest parentage. (**B**,**C**): Representation of the distribution by country for Africa and Europe. (**D**): Country distribution for North America, Asia and Australia.

**Figure 3 ijerph-22-00770-f003:**
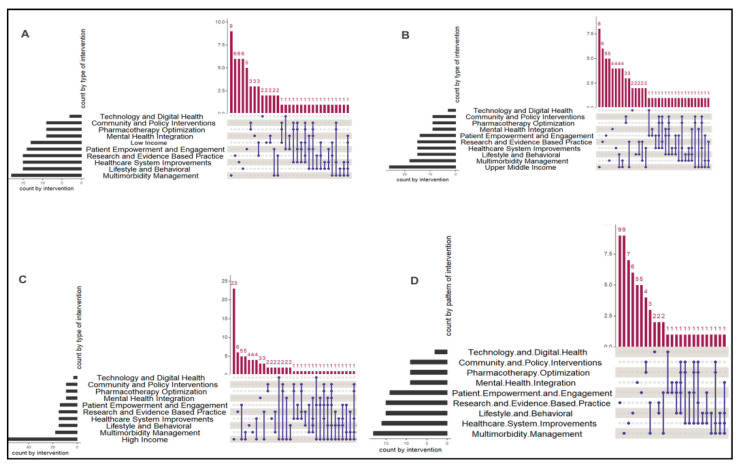
Distribution of interventions in the 101 studies included in the review, stratified by countries’ financial status. (**A**) Low income; (**B**) upper-middle income; (**C**) high income; (**D**) global. upSet plot generated using Rstudio. Studies included recommended intervention, which formed patterns. Red vertical bars represent the number of studies that included a specific intervention combination. The blue dots and connecting lines in the matrix below the red bars indicate which intervention types are present in each combination. A blue dot without a connecting line represents a stand-alone combination. The black horizontal bars represent the total number of studies that included each intervention type, regardless of whether it was combined with others.

**Figure 4 ijerph-22-00770-f004:**
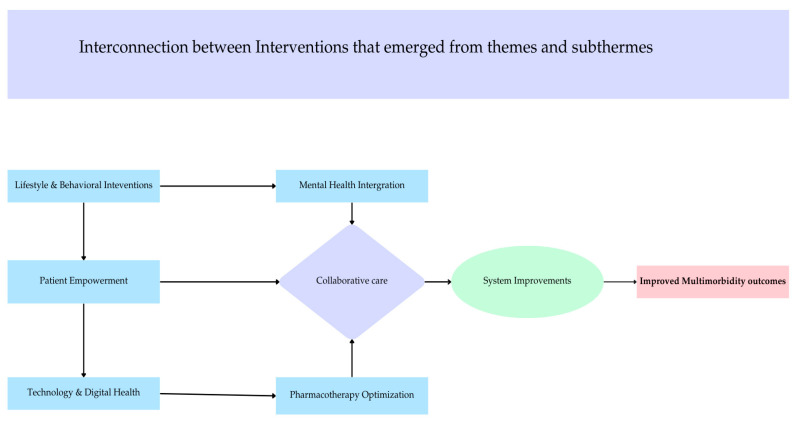
The inter-relatability of themes that emerged.

## Data Availability

All data produced in the present work are contained in the manuscript.

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
