# Peer review of "Multimorbidity Management: A Scoping Review of Interventions and Health Outcomes"

_ijerph, 2025, doi:10.3390/ijerph22050770_

Round 1

Reviewer 1 Report

Comments and Suggestions for Authors

As per the attached PDF review report.

Comments on the Quality of English Language

As per the attached PDF review report.

Author Response

The responses to reviewers comments are attached. 

Reviewer 2 Report

Comments and Suggestions for Authors

The manuscript presents a well-structured scoping review that synthesizes existing interventions for multimorbidity management. Given the increasing burden of multimorbidity worldwide, the study is timely and relevant. It successfully compiles and categorizes interventions, including pharmacological treatments, lifestyle modifications, coordinated care models, and digital health innovations. Following the PRISMA-ScR framework, the methodological rigor strengthens the review’s credibility.

Multimorbidity, defined as the co-occurrence of two or more chronic conditions, presents a significant challenge to healthcare systems globally. This scoping review analyzes 101 studies published between 2012 and 2024, including over 1.2 million individuals. The findings highlight the importance of multidisciplinary interventions, patient empowerment strategies, and the integration of digital health tools. Key themes include lifestyle and behavioral interventions, health system improvements, and optimization of pharmacotherapy. The study underscores the necessity of holistic, patient-centered approaches to managing multimorbidity effectively.

Here are some suggestions:

  • The manuscript uses "multimorbidity interventions" and "comprehensive interventions" interchangeably. It would be beneficial to define these terms explicitly and ensure consistency throughout the text to enhance clarity.
  • The manuscript states that 16.2% of participants were male, while 83.8% were female. The significant gender disparity warrants further discussion—was this an expected distribution based on the literature, or does it indicate a sampling bias?
  • The manuscript briefly mentions that socioeconomic status may impact the effectiveness of interventions. Expanding on this would strengthen the discussion.
  • Figures and tables provide valuable data visualization, but some graphs could benefit from more precise labeling and captions. For instance, Figure 2 could explicitly state whether the distribution of studies reflects global multimorbidity prevalence.

Author Response

The responses to reviewer's comments are attached. 

Reviewer 3 Report

Comments and Suggestions for Authors

Dear Editor, 

Thank you for the invitation to review this manuscript. This manuscript is a scoping review of comprehensive interventions for multimorbidity management. While the topic is interesting and might be beneficial for the body of knowledge in this field, there are some concerns regarding the manuscript that need extra attention from the authors. Below are our comments and suggestions for the authors.

Title:

1) Please remove the period after the title. A title is not a complete sentence; therefore, it does not need a period at the end of it.

Abstract:

2) No need to provide global data prevalence in the abstract. When providing the percentage prevalence, references are needed. However, the abstract should not contain any references. Therefore, avoid presenting detailed information, especially data in percentage, in the introduction of the abstract.

Introduction:

3) Clear

Methods:

4) On page 3, lines 93–94, it is written that “The search strategy for PubMed has been included in the supplementary file as an example.” However, in the supplementary file, the search strategy for PubMed cannot be found. Please add.

5) On page 3, lines 134–135, it is written that Figure 3 presents the distribution of interventions. However, figure 3 is not found in the manuscript. The manuscript only contains 2 figures. Please clarify.

6) Table 1 is too long. It takes 30 pages (from page 6 to page 36) to present only Table 1. Instead of presenting Table 1 in the manuscript, I suggest the authors move it as a supplementary file.

7) In Table 1, there is a column for “Female %,” and in that column, there are numbers, such as 2017, 2620, etc. What is that column for? If it is for the percentage of female subjects, how can the percentage reach 2017, 2620? Please clarify.

8) Information presented in Table 1 is not complete. Some information, such as type of study and sample size, is missing in some studies, such as by Adam et al. (2019), Afshar et al. (2015), Lin et al. (2014), etc. Please complete the information.

9) The intervention column in Table 1 should contain interventions from the reviewed studies that are related to the objectives of this systematic review. Therefore, suggestions such as “Future intervention studies with long follow-up periods are warranted (page 22),” or “Future research, preferably longitudinal design, should investigate the possible causal underlying mechanism…” (page 24), or “Future research to better understand the relationship between…” (page 25) are not necessary to be presented in the table because they do not contain information regarding interventions. In this current version of Table 1, interventions for multimorbidity outcomes, based on the objectives of this study, are not clearly presented.

10) More information about data analysis is needed. On page 3, lines 116–118, the authors mentioned that relevant data were extracted from the articles using Microsoft Excel, and the extracted data included author name, date of publication, etc. Then, in the results, the authors identified 9 themes and sub-themes. How did these themes and sub-themes emerge? What process did the authors undertake in the analysis in order to derive these themes and sub-themes? Please explain.

Results:

11) Findings revealed nine themes. However, there is no explanation about each theme. I suggest the authors elaborate the description of each theme before explaining sub-themes.

12) One of the themes is “multimorbidity management.”. Isn’t this manuscript all about interventions for multimorbidity management? Naming a theme as “multimorbidity management” implies that multimorbidity management is a part of the study, while in fact, multimorbidity management is about the whole of the manuscript. Instead of using multimorbidity management as a theme name, I suggest the authors change the name of that theme based on the sub-themes establishing that theme.

Discussion:

13) Please add strengths and limitations of this study.

Conclusion:

14) Clear

Comments on the Quality of English Language

Quality of English language for the whole manuscript needs to be improved. There are some typos and grammatical errors. Some sentences are not well structured and difficult to understand. For example:

1) On page 4, figure 2, it is written “Ashia.” Does that refer to a continent? If yes, it should be “Asia,” not “Ashia.” Also, please check other typos: “country districution” and “continants.”

2) On page 37, it is written, “Lifestyle interventions such as quitting smoking dietary advice reducing alcohol consumption we found in several studies to have a positive impact on the management multimorbidity.” This sentence is not effective and difficult to understand. This sentence should be revised to be more effective, such as: “Lifestyle interventions such as quitting smoking, healthy diet, and alcohol reduction were found to have a positive impact on the management of multimorbidity.”

3) On page 38, lines 189–191, it is written, “In addition, addressing key non-disease-specific self-management challenges and themes (mastering emotions, managing treatments, and communication within health care).” This is not a complete sentence.

4) On page 38, lines 191–192, it is written, “Developing accessible educational resources for patients on various health topics.” This is not a complete sentence. Please revise and write clearly what the authors meant to say.

5) On page 39, lines 221–224, it is written, “Focusing on the individual needs and preferences of patients, implementing personalized care plans that address multiple health conditions, restructuring healthcare delivery models to be more efficient and patient-centred and implementing care teams and case management approaches.” This sentence is too long, difficult to understand, and missing the structures of a complete sentence. What are the subject and predicate of this sentence? What the authors intended to say is not clear from this sentence.

Author Response

(The authors gave the same response as above.)

Reviewer 4 Report

Comments and Suggestions for Authors

This scoping review explores various interventions for managing multimorbidity, highlighting strategies such as lifestyle modifications, pharmacotherapy optimization, care coordination models, and digital health innovations. The manuscript is well-written and provides a comprehensive synthesis of existing literature. I have a few minor comments for the authors' consideration to enhance clarity and consistency.

  1. Lines 123–124: The authors state that the final analysis included 101 articles selected from 3,119 published between 2012 and 2024. However, the inclusion criteria do not specify the publication time frame for the selected articles. If a publication date restriction was applied, please clarify this in the inclusion criteria. If not, please explain why articles published before 2012 were excluded.
  2. Figure 1: There is one asterisk after “Records identified from” and two asterisks after “Records excluded,” but no explanation is provided in the figure footnotes. Please add appropriate footnotes to clarify their meaning.
  3. Lines 165–167: This sentence is unclear and appears incomplete. Please rephrase it for better readability and precision.
  4. Theme and Subtheme Numbering: I recommend numbering the themes and subthemes in the text to align with Table 2. While the authors use italics for themes, some subthemes are also italicized (e.g., Assisting patients with goal setting and prioritization, peer support, health services navigation), which may cause confusion. Numbering would help readers more easily distinguish between themes and subthemes.

Author Response

(The authors gave the same response as above.)

Reviewer 5 Report

Comments and Suggestions for Authors

The paper provides a systematic review of multimorbidity management interventions, analyzing 101 studies and identifying nine key intervention themes. Each theme is well-reviewed and discussed. The paper is well-structured, clearly written, and supported by evidence.

Here are some suggestions and questions for the authors:

  1. Suggestions for Improving Clarity and Informativeness in Table 1:
    a. The title of Table 1 is currently read "Table 1. Table S1: Summary of included studies." Please remove "Table S1" for consistency.
    b. In the "Female%" column, absolute numbers are provided instead of proportions. Please adjust either the column title or the values to ensure clarity.
    c. Several cells in the table are left empty. Consider filling them with placeholders such as "Not provided" or "NA" to maintain consistency.
    d. The table does not clearly indicate what outcomes are measured for each intervention. Consider adding an "Outcome" column to briefly specify the measured outcomes or include a brief discussion highlighting common outcomes assessed across interventions.
  2. In Table 2, the authors identify nine themes of interventions. However, it is not immediately clear how these themes are linked to the studies presented in Table 1. To enhance clarity and usability, consider adding category labels in Table 1 or providing a direct reference to Table 2 for easier cross-referencing.
  3. The authors clearly review and discuss each intervention individually. Additionally, could the authors explore the potential interconnections between the interventions? Understanding these relationships could provide valuable insights into their combined effects. A graphical representation of these connections would also be helpful.
  4. Regarding the implementation of the interventions, could the authors summarize or discuss the challenges associated with these interventions, as well as the level of effort required for their effective implementation?

Author Response

(The authors gave the same response as above.)

Round 2

Reviewer 3 Report

Comments and Suggestions for Authors

Dear Editor,

Thank you for the opportunity to review the revised version of the manuscript. I appreciate the authors’ effort to revise the manuscript. However, after reviewing the revised version, the manuscript did not change significantly, and there are more parts that need clarifications. Please find the detailed comments below.

Abstract:

  • It is correct that in the previous version I suggested the authors remove the details of prevalence percentage information in the background of the abstract, but it did not mean that they just remove the number without revising the context of the sentence. Now the sentence is read as “With a prevalence globally, this complex health phenomenon is increasingly affecting populations as they age,” which does not sound complete.
  • In the abstract, it is written that this review included 101 studies, but in Table S1, only 100 studies. One study is missing.

Materials and methods:

  • As the authors mentioned following the PRISMA-ScR checklist in the previous version, but the checklist was not found, I suggested the authors attach the checklist. In this revision, the authors indeed provided a PRISMA-ScR checklist in the supplementary file, but it was in the form of an empty checklist without any information on which page each item is located. Please understand that when the authors stated that they followed the protocol based on PRISMA-ScR checklist and provided the checklist, the readers will expect it is the filled in checklist where the information of each item can be located, not an empty checklist.
  • This study aims to review interventions to manage multimorbidity, and for this aim, the authors included 101 studies. However, the derived information in some included studies presented in the table is very superficial and did not show the intervention needed for this review. For example, study By Romano et al. (2021), in the intervention column, it is written that “future longitudinal research is required to assess the impact of dietary reduction on multimorbidity incidence.” This statement is not the interventions that are reviewed from the included study.
  • Another example is the study by Roomaney et al. (2022), in the intervention column, it is written that, “More studies are needed to identify common disease clusters and multimorbidity trends to assist in the endeavour of targeting high-risk people.” This statement does not inform the intervention.
  • Another example is the study by Roomaney et al. (2023). In the intervention column, it is written that, “A need for hypertension to be addressed.” This sentence is not clear, especially in relation with the intervention that is needed in this study.

Results: 

  • The authors wrote that the total number of participants was 1,446,304, with 199,413 being male and 1,028,493 being female. This calculation is not correct. The total number of males and females based on those numbers of participants should be 1,227,906. Also, in Table S1, some studies do not have information regarding the number of female participants. So, how did the authors calculate the total number of female participants then?
  • Also, the authors mentioned the distribution of studies based on geographical locations and the total study count for 98 studies (27 in Africa, 5 in Asia, 2 in Australia, 30 in Europe, 15 in North America, and 19 multinational). This information is missing 3 studies if the total number of studies is 101 studies.
  • In the previous comments, I suggested the authors correct some typos. However, typos are still found, such as “continants” still being written as it is; it is not revised to “continents.”
  • In Figure 2, the percentage of studies from India has changed from the previous version. In the previous figure, studies from India count for 40%, and in the recent figure, studies from India count for 4.95%.
  • On page 4, line 146, it is written that there are two studies from Australia. However, in Figure 2, studies from Australia count for 100. It is not clear whether 100 is a percentage or a frequency. Nonetheless, either percentage or frequency, the value 100 does not make sense for studies that count only 2 from 101 studies. What does 100 mean?  

Supplementary file:

  • There is no legend for the supplementary table. In the female column, NR is written frequently, but there is no information on what NR stands for.
  • In the previous comments, I suggested the authors complete missing information in the table. However, there is still some missing information in the table S1, such as the study by Roche et al. (2017) is missing information regarding participants’ characteristics, the study by Salari et al. (2022) is missing the study type, and the study by Widmann et al. (2017) is missing the information about female participants.
  • The study by Freisling et al. (2020) needs more elaboration regarding the intervention. The authors wrote in the intervention column, “Healthy lifestyles prior to reduce risk of multimorbidity.” What does this statement mean? What lifestyle reduces the risk of multimorbidity? 
  • Study by Hien et al. (2014): In the intervention column, it is written that “Research to develop innovative interventions to reduce the burden of multimorbidity in sub-Saharan Africa.” This statement needs elaboration. It is not clear what this statement means.
  • Study by Hirst et al. (2021), the statement, “Regular blood pressure monitoring to guide treatment to optimal blood pressure targets with antihypertensive.” This sentence is difficult to understand. Please paraphrase and make it clear what this means.
  • The study by Lear et al. (2021) needs more information on the intervention column.
  • Study by Mbokazi et al. (2023): The statement in the intervention column needs to be paraphrased. In that column, the authors wrote, “We decolonize individual illness behavior theory developed in HICs and move towards acknowledging culturally embedded patterns of support and collaboration.” Since the intervention is not established by the authors but by Mbokazi et al., the word “we” to describe the intervention is not appropriate.
  • Information in the intervention column is difficult to understand. I trust the authors derived the information from the articles; however, when putting it into the table as a sentence, the information is difficult to read. For particular intervention, the authors should list the intervention so it will read as points, not as a continuous sentence.  
  • In the study by Odland et al. (2022), in the intervention column, it is written, “Politicians and policymakers in low-income countries, and global health funding, need to focus on chronic conditions.” It is not clear what the interventions are in this study.
  • The authors mentioned in the limitation that the included studies consisted mostly of females compared to males. How do the authors know that the female participants count more than the male participants in the included studies, while in Table S1, some studies do not have information on the number of the female participants, such as studies by Adam et al. (2019), Afshar et al. (2015), Blum et al. (2013), Coventry et al. (2015), Khunti et al. (2021), etc.?

Comments on the Quality of English Language

The English writing is difficult to understand.

Author Response

(The authors gave the same response as above.)

Reviewer 5 Report

Comments and Suggestions for Authors

I sincerely appreciate the authors' efforts in addressing the questions. While there are notable improvements, with some concerns effectively addressed in the text, others may still require further clarification.

As a general suggestion, the response letter would be more informative if it provided specific references to the revised text rather than simply stating 'Corrected'. Indicating the exact lines/sections where changes were made, along with brief explanations or context, would help better convey the authors' rationale and insights.

Specifically, I have the following questions:

  1. Regarding my earlier comment: 'In Table 2, the authors identify nine themes of interventions. However, it is not immediately clear how these themes are linked to the studies presented in Table 1. To enhance clarity and usability, consider adding category labels in Table 1 or providing a direct reference to Table 2 for easier cross-referencing'. The response states that this has been 'Corrected'. I understand that the relevant tables are now Table 1 and Table S1. However, it remains unclear how the 101 studies in Table S1 align with the categories in Table 2. Could the authors clarify how this has been addressed?
  2. I appreciate the inclusion of Figure 3 to visualize the distribution of interventions across the reviewed studies. While some elements, such as 'count by intervention', are straightforward, others—including the red bar at the top and the blue dots and lines at the bottom—are difficult to interpret. Adding descriptions to the legend would help clarify these elements for readers. Additionally, I do not see any reference to Figure 3 in the main text. Please ensure that it is cited appropriately and provide relevant discussion to enhance its interpretability.
  3. The addition of Figure 4, illustrating the interconnections between interventions, is a valuable enhancement. Please ensure that this figure is also properly referenced in the main text and accompanied by relevant discussion.

Author Response

(The authors gave the same response as above.)
